

# Impact of mini-driver genes in the prognosis and tumor features of colorectal cancer samples: a novel perspective to support current biomarkers

Anthony Vladimir Campos Segura[1,2,3], Mariana Belén Velásquez Sotomayor[3,4], Ana Isabel Flor Gutiérrez Román[1,2], César Alexander Ortiz Rojas[3,5] and Alexis Germán Murillo Carrasco[3,6]

[1] Biochemistry and Molecular Biology Research Laboratory. Faculty of Natural Sciences and Mathematics, Universidad Nacional Federico Villarreal, Lima, Peru
[2] Research Group in Biochemistry and Synthetic Biology (GIBBS-UNFV), Lima, Peru
[3] Research Group in Immunology and Cancer (IMMUCA), Lima, Peru
[4] School of Human Medicine, Faculty of Health Sciences, Universidad Científica del Sur, Lima, Peru
[5] Hematology Division, LIM31, Medical School, Universidade de São Paulo, Sao Paulo, Brazil
[6] Centro de Investigação Translacional em Oncologia (LIM24), Departamento de Radiologia e Oncologia, Faculdade de Medicina, Universidade de São Paulo, Sao Paulo, Brazil

Corresponding author
Alexis Germán Murillo Carrasco, agmurilloc@usp.br

## ABSTRACT

**Background:** Colorectal cancer (CRC) is the second leading cause of cancer-related deaths, and its development is associated with the gains and/or losses of genetic material, which leads to the emergence of main driver genes with higher mutational frequency. In addition, there are other genes with mutations that have weak tumor-promoting effects, known as mini-drivers, which could aggravate the development of oncogenesis when they occur together. The aim of our work was to use computer analysis to explore the survival impact, frequency, and incidence of mutations of possible mini-driver genes to be used for the prognosis of CRC.

**Methods:** We retrieved data from three sources of CRC samples using the cBioPortal platform and analyzed the mutational frequency to exclude genes with driver features and those mutated in less than 5% of the original cohort. We also observed that the mutational profile of these mini-driver candidates is associated with variations in the expression levels. The candidate genes obtained were subjected to Kaplan–Meier curve analysis, making a comparison between mutated and wild-type samples for each gene using a $p$-value threshold of 0.01.

**Results:** After gene filtering by mutational frequency, we obtained 159 genes of which 60 were associated with a high accumulation of total somatic mutations with $Log_2$ (fold change) > 2 and $p$ values < $10^{-5}$. In addition, these genes were enriched to oncogenic pathways such as epithelium-mesenchymal transition, hsa-miR-218-5p downregulation, and extracellular matrix organization. Our analysis identified five genes with possible implications as mini-drivers: *DOCK3, FN1, PAPPA2, DNAH11*, and *FBN2*. Furthermore, we evaluated a combined classification where CRC patients with at least one mutation in any of these genes were separated from the main cohort obtaining a $p$-value < 0.001 in the evaluation of CRC prognosis.

**PeerJ** ________________________________

**Conclusion:** Our study suggests that the identification and incorporation of mini-driver genes in addition to known driver genes could enhance the accuracy of prognostic biomarkers for CRC.

## INTRODUCTION

Colorectal cancer (CRC) is a significant global health problem, ranking as the third most diagnosed neoplasia and the second leading cause of cancer-related death worldwide (*Bray et al., 2018*; *Sung et al., 2021*). In 2020, there were approximately 1.93 million new cases of CRC, representing 10% of all cancer cases (*Sung et al., 2021*). Despite advances in detection and treatment, CRC incidence and mortality rates have increased by 7.2% from 2018 to 2020. Therefore, it is essential to continue research into the underlying mechanisms driving CRC progression (*Bray et al., 2018*; *Sung et al., 2021*).

*KRAS*, *NRAS*, *BRAF*, *TP53*, and *APC* are considered driver genes for CRC since a few pathogenic mutations in any of these genes are sufficient to develop a tumor (*Thierry et al., 2014*). However, defining a single group of genes as the drivers of all tumors is challenging and only explains a small number of cancer cases (*Thierry et al., 2014*). This challenge arises because the description of any genetic variation as a pathogenic mutation depends on factors such as its impact on the translated protein (missense, nonsense, *etc.*) or the number of nucleotides affected (single-nucleotide, insertion, deletion, *etc.*).

To differentiate deleterious variations from polymorphisms, current genetic definitions consider mutations to be any genetic variations with a reduced frequency (<1%) in a healthy population (*Al-Koofee & Mubarak, 2020*). However, these definitions do not account for the possibility of new transcriptional switches that can support novel consequences of previously known polymorphisms, as recently described in CRC (*Abdi, Latifi-Navid & Latifi-Navid, 2022*).

The identification of coding genes, lncRNA, circRNA, and miRNA through transcriptomic studies has broadened the concept of mini-driver genes to genes or genomic regions that may collectively be associated with poor prognosis in cancer (*Yang et al., 2018*; *Wu et al., 2020*). Nevertheless, considering the large group of genes associated with cancer features and prognosis whose function is not elucidated, it is essential to propose a strategy that includes all types of genetic variations as mutations for analyzing candidate genes capable of supporting current driver genes. To this end, we adopt the concept of mini-driver genes, which refers to low-frequency genetic alterations with a relatively weak tumor-promoting effect (*Castro-Giner, Ratcliffe & Tomlinson, 2015*).

There are established criteria for identifying mutated genes that may be considered mini-driver genes. Firstly, individual mutations in a mini-driver gene should provide a growth advantage for cancer cells compared to normal cells, although this is not necessarily critical for tumor development. Secondly, mini-driver genes must be present in

a small proportion of tumors. Thirdly, they may be present in subclones because they have a relatively weak selective advantage, resulting in less probability of selective sweeping. Fourthly, mini-driver genes should show parallel or convergent evolution between cancer subclones and between cancers of the same type. Finally, they should be involved in processes such as gene expression regulation, mRNA stability, transcriptional changes, DNA methylation, and other non-coding genomic features (*Castro-Giner, Ratcliffe & Tomlinson, 2015*; *van Ginkel, Tomlinson & Soriano, 2023*).

Mini-driver genes play a significant role in tumor diversification, but the mechanisms underlying their effects are not well understood. One possible mechanism is that mini-drivers may help to maintain tumor homeostasis by counteracting the deleterious effects of some passenger mutations (*Li & Thirumalai, 2016*; *Cuykendall, Rubin & Khurana, 2017*). In some cases, non-coding mutations could be called "mini-drivers" because they alter transcriptional regulation, mRNA translation and stability, splicing control, and chromatin structure, leading to altered gene expression that favors tumor progression (*Elliott & Larsson, 2021*). Another possible mechanism by which mini-driver genes contribute to tumor development is through working together with driver genes/ mutations. Large-scale genome-wide association studies (GWAS) have shown that even the most significant loci explain only a fraction of the predicted genetic variation for typical traits (*Boyle, Li & Pritchard, 2017*). Therefore, mini-driver genes may explain how polygenic effects provide a means by which heterogeneous mutation patterns can generate distinctive changes consistent with the phenotype observed in tumors (*Bennett et al., 2018*).

Clinical studies have identified mini-driver genes as prognostic biomarkers in cancer (*Bennett et al., 2018*) and our group evaluated mini-driver features in selected genes (*Campos Segura, 2022*) to focus on their contribution to CRC progression. In this study, we propose a strategy to identify potential mini-driver genes in CRC using Next Generation Sequencing (NGS) data and determine whether they could serve as prognostic markers, providing insights into their role in CRC progression.

## MATERIALS AND METHODS

### Database filtering and mutational frequency analysis

We retrieved data from Next Generation Sequencing (NGS) experiments and clinical information of colorectal cancer (CRC) patients using the cBioPortal platform (https://www.cbioportal.org/) ("cBioPortal for Cancer Genomics"; *Cerami et al., 2012*). Genomic data was only utilized to provide mutational information, while transcriptomic data, when available, was used to obtain mutational status, prediction of copy number alterations, and gene expression levels. Three cohorts of colorectal adenocarcinoma were selected for analysis, including the Dana-Farber Cancer Institute (DFCI) cohort ($n = 619$) (*Giannakis et al., 2016*), the Pan-Cancer Atlas from The Cancer Genome Atlas (TCGA) cohort ($n = 594$) (*Liu et al., 2018*), and the Memorial Sloan Kettering—Cancer Center (MSKCC) cohort ($n = 138$) (*Brannon et al., 2014*) datasets. All data sets were generated using Illumina HiSeq sequencers, including information for 18,215 genes. Clinical data are summarized in Table 1.

**Table 1 Clinical and biological characteristics in colorectal cancer.**

| Variable | | Study participants ($n$ = 1,351) |
| --- | --- | --- |
| Age (average ± standard deviation) | | 68.42 ± 11.14 |
| Sex, $n$ (%) | Female | 660 (48.9) |
| | Male | 551 (40.8) |
| | N/A | 140 (10.4) |
| CIMP category, $n$ (%) | CIMP-high | 95 (7.0) |
| | CIMP-0/low | 405 (30.0) |
| | N/A | 851 (63.0) |
| MSI status, $n$ (%) | MSI-high | 91 (6.8) |
| | MSS | 438 (32.4) |
| | N/A | 822 (60.8) |
| Tumor stage, $n$ (%) | I | 152 (11.3) |
| | II | 187 (13.8) |
| | III | 159 (11.8) |
| | IV | 65 (4.8) |
| | N/A | 788 (58.3) |
| Tumor site, $n$ (%) | Rectum | 137 (10.2) |
| | Cecum | 114 (8.4) |
| | Ascending to transverse colon | 201 (14.8) |
| | Splenic flexure to sigmoid colon | 166 (12.3) |
| | N/A | 733 (54.3) |

**Note:**
N/A, No Available; CIMP, CpG island methylator phenotype; MSI, Microsatellite Instability.

## Characterization of somatic mutational profile in CRC

The cBioPortal platform was used to generate a report that displays the number of genetic variants detected in each gene, the number of patients with at least one variant per gene, and the total number of patients with available mutational information per gene. For this study, we included all somatic genetic variations, encompassing both SNPs (≥1% frequency in populations) and pathological mutations (<1% frequency in populations), as mutations. We then calculated the mean number of mutations per gene and patient (Table S1).

Using these values, we established four groups of genes. For that, we consider the power of the sample (number of patients), the percentage of participating genes, and the ranking of common driver genes in CRC. As result, we classified the genes as rarely mutated (<7%), low-mutated levels (7–10%), moderately mutated (11–50%), and highly mutated (>50%) genes. To visualize these results, we designed a scatterplot using the R software v.4.2.0 (R Core Team, 2022) with the ggplot2 package (Wickham, 2011).

## Association between mutational status and tumor mutational burden

After filtering the prior data, we selected a putative group carrying mini-driver genes, consisting of low-mutated genes in patients (7% and 10%). Based on these genes, we compared the total number of variations (tumor mutational burden, TMB) per patient

according to their mutational status per gene. In our comparison, we considered any patient with at least one somatic variation in a specific gene as "mutated". To compare TMB, we utilized the Mann–Whitney test and calculated the fold change between mutated and wild-type groups per gene. We then adjusted the *p*-values using the Benjamini–Hochberg (BH) method (*Benjamini & Hochberg, 1995*). Finally, we established a mean-value threshold ($10^{-5}$) for selecting the top genes whose mutational status was associated with changes in TMB. To visualize our findings, we plotted the results using the Enhanced Volcano package in the R software ("EnhancedVolcano"; *Blighe, Rana & Lewis 2018*).

## Association between mutational status and gene expression/copy number variations

We investigated the expression levels of genes with mutational status and copy number variations. To accomplish this, we used the TCGAretriever package ("TCGAretriever: Retrieve Genomic and Clinical Data from TCGA"; *Fantini, 2019*) to verify the normalized expression levels of genes and putative copy number alterations predicted by Gistic (*Mermel et al., 2011*) in colon and rectal adenocarcinoma samples. We applied the same criteria as in the previous step to dichotomize the patient population for each selected gene (mutated or wild-type) and compared the expression levels of genes or copy number alterations. We then used the Mann–Whitney test to compare these values, taking into account the distribution of expression levels. Finally, we represented all samples in boxplots for each pair of genes. The Y-axis displays expression levels or the number of altered copies of the first gene, while the X-axis shows patient categories based on the mutational status of the second gene.

## Gene enrichment

Gene lists were obtained after applying the Enrichr tool (*Xie et al., 2021*). We used all available databases from Transcription (17 databases), Ontology (25 databases), and Pathway (eight pathways) modules. After loading gene lists, we only considered relevant pathways where more than three genes were associated with $p < 0.001$.

## Survival analysis

To investigate the prognostic value of the mini-drivers, we employed Kaplan–Meier curves and Cox proportional hazard regression analysis (*Borgan Rnulf, 2001*). Then, we identified the mini-drivers that potentially play a role in disease progression, including *DOCK3*, *FN1*, *PAPPA2*, *DNAH11*, and *FBN2*. Then, we established a gene panel based on these genes. Cox regression analysis was performed, including age (as continuous), sex, and tumor stages (AJCC, American Joint Committee on Cancer) as potential confounding variables. To perform these analyses, we utilized the survival and survminer packages of R software version 4.2.0 (*R Core Team, 2022*).

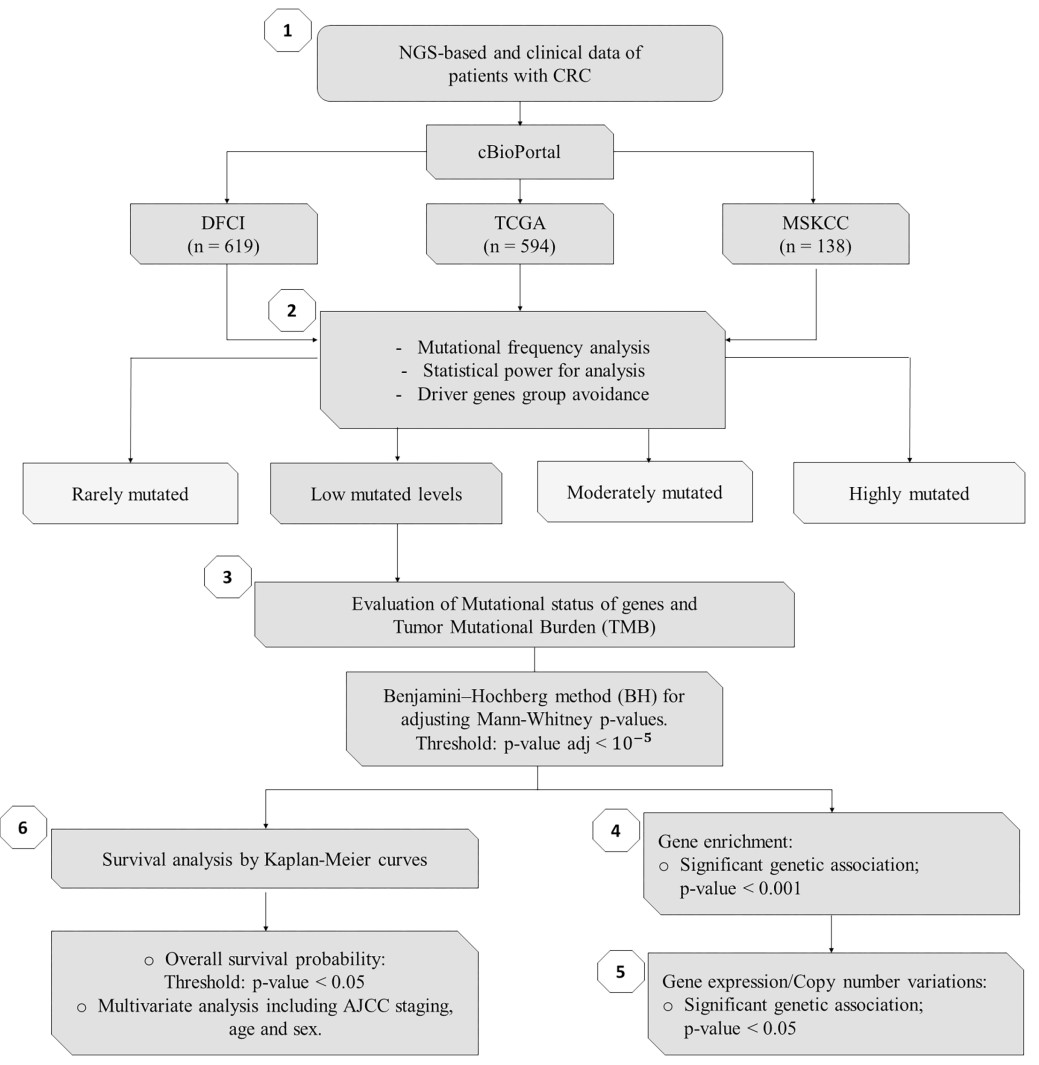

**Figure 1 Flowchart of methods of this study.** NGS, Next-Generation Sequencing; CRC, Colorectal Cancer; DFCI, Dana-Farber Cancer Institute; TCGA, The Cancer Genome Atlas; MSKCC, Memorial Sloan Kettering Cancer Center; *n*, number of samples.

## RESULTS

### Identifying mini-drivers in CRC based on the mutational frequency

As summarized in Fig. 1, to identify potential mini-drivers in CRC, we analyzed the gene mutation frequency, and the statistical power. Then, we classified the genes into four groups (Table S1, Fig. 2): rarely mutated (≤7%, 17,993 genes), lowly mutated (7–10%, 159 genes), moderately mutated (11–50%, 62 genes), and highly mutated (>50%, two genes). The rarely mutated group includes genes with less than one variation in at least 59 patients, which represents less than 5% of the patients with available information for mutational profiling (980 patients from three cohorts with available information). To avoid non-representative results in further analyses, we excluded this group of genes.

The moderately and highly mutated groups include genes considered as drivers, such as *APC* with 66%, *TP53* with 56%, *KRAS* with 35%, *BRAF* with 16%, and *ARID1A* with 11%,

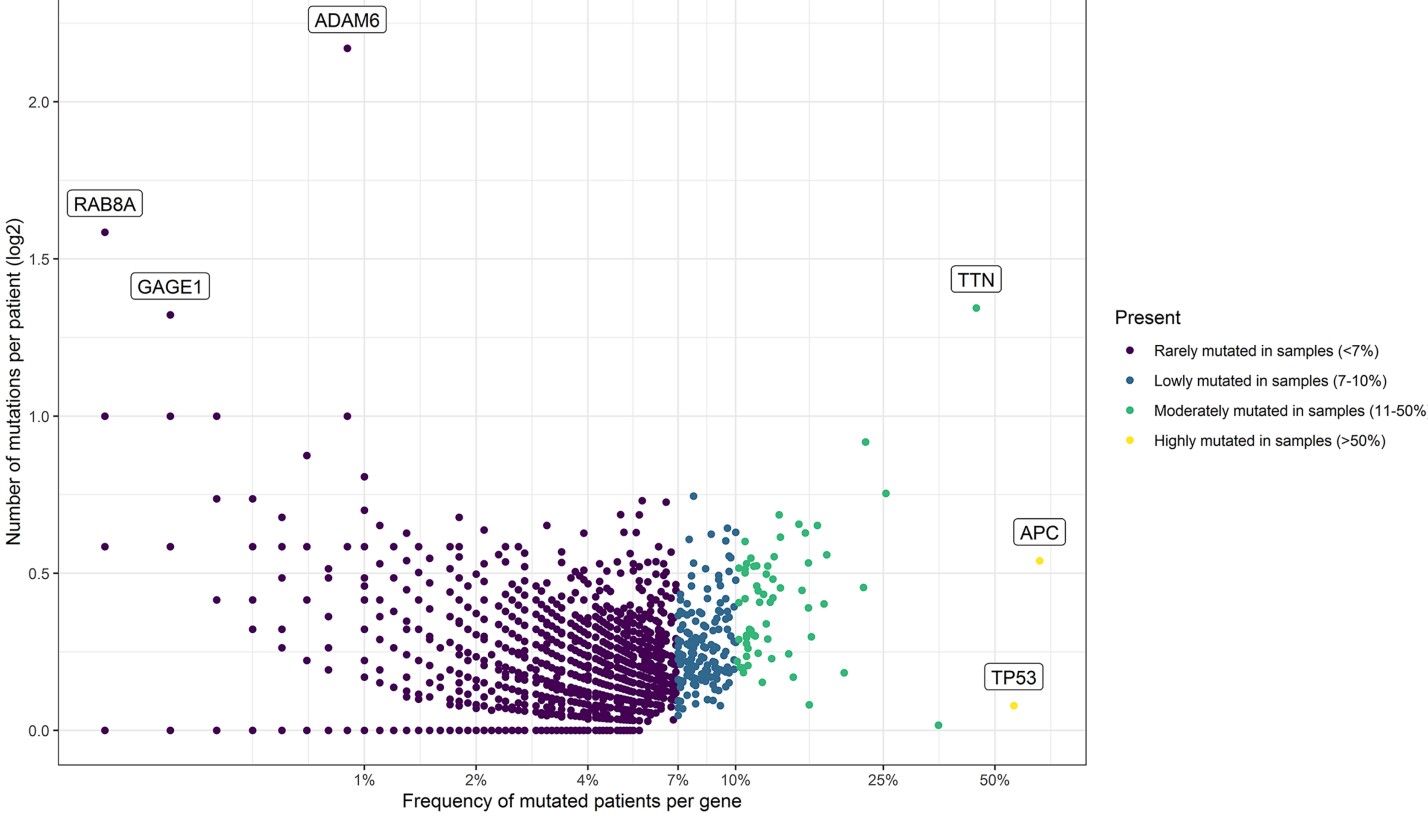

**Figure 2 Mutational frequency present in the samples obtained from the three cohorts: DCFI, TCGA, and MSKCC.** Based on the number of mutations per patient we classify the mutated genes into four groups: rarely mutated (<7%), low mutated levels (7–10%), moderately mutated (11–50%), and highly mutated (>50%). Name labels were added to recognize prominent genes with a high number of mutations per patient or a high percentage of affected patients.

among others. We focused on exploring the genes in the lowly mutated group (159 genes) as potential mini-drivers. We analyzed these genes with the Enrichr database and found that they are mainly associated with hsa-miR-218-5p regulation (miRTarBase 2017 database, adjusted $p$-value = $1.9 \times 10^{-4}$), extracellular matrix organization (Reactome 2022 database, adjusted $p$-value = $9.1 \times 10^{-11}$), and epithelial-mesenchymal transition (EMT, MsigDB Hallmark 2020 database, adjusted $p$-value = $2.5 \times 10^{-8}$).

## Putative mini-driver genes are associated with high mutation rates

In order to assess the impact of mini-drivers on tumoral progression, we analyzed the polygenic effect of 159 genes selected from the previous step. Our goal was to identify which of these genes were associated with high mutation rates. As shown in Fig. 3, we found that 60 genes had a significantly higher TMB when mutated compared to their respective wild-type group, with an increase ranging from 5.4–8.7-fold ($p$-value < $10^{-5}$). Among the most statistically significant genes associated with high TMB were *MUC5B*, *DNAH7*, *DOCK3*, and *BMPR2*. Meanwhile, we identified *SYNE2*, *COL7A1*, *NOTCH3*, and *SPEG* as the genes with the highest mutation counts.

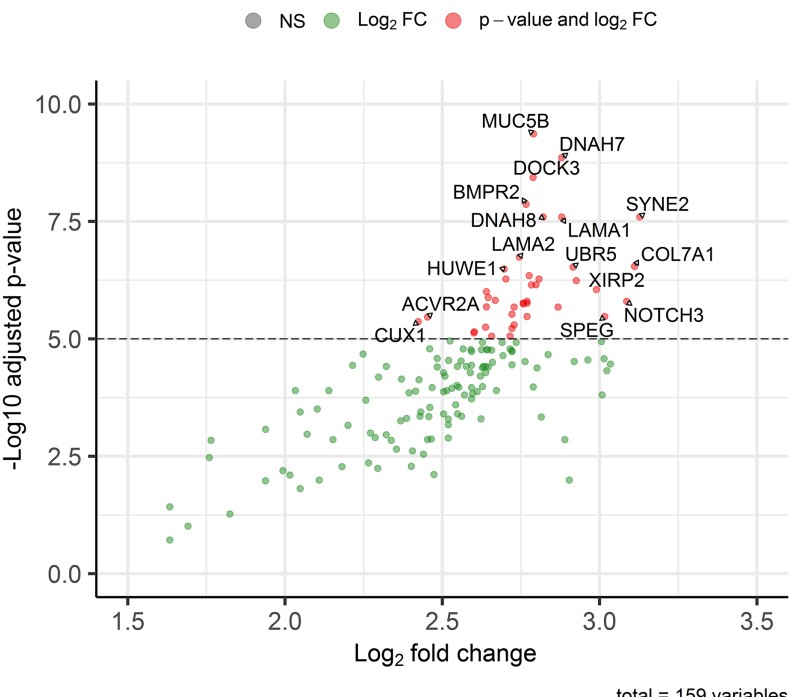

**Fold change of mutation count classified by specific genes**

p values were obtained using the Mann-Whitney test || pCutoff=1e-5

**Figure 3** Association of the mutational status of genes and the total mutation count in CRC patients. Scatter plot showing $\log_2$Fold Change and adjusted $p$-values between the number of mutations in patients classified as mutated or wild-type for each gene. The $p$-value threshold was set at $10^{-5}$ to show strongly associated genes with high mutation rates in CRC patients.

## Mini-drivers are associated with specific gene signatures in CRC

After identifying the genes associated with high TMB (Fig. 3), we compared the expression levels and copy number alterations with the mutational status of these genes.

We discovered 46 genes whose mutational status was linked to changes in the expression levels of other genes ($p < 0.05$, Table S2). Remarkably, the mutated group of the *BMPR2* gene alters the expression of nine genes (*ACVR2A, DOCK3, MUC5B, FLNA, DNAH8, SYNE2, FBN2, ITPR3,* and *DLC1*), whereas mutated groups of *FLNB, ACVR2A, SIPA1L3,* and *CELSR1* genes altered the levels of other six genes. Table 2 provides a summary of the 43 relevant gene pairs ($\log_{10}(FC) > 0.4$, $p < 0.01$), while Fig. 4 displays a selection of these findings. Among them, we found that the subexpression of *MYH11* was related to its mutational profile ($p = 0.0043$), *UBR5* levels were decreased in patients with mutations in *FLNB* ($p = 0.0012$) or *COL7A1* ($p = 0.0075$), whereas *FBN3* mutated was associated with low expression of *FBN2* ($p = 0.0079$). Additionally, we observed reduced expression levels of *ZNF536* and *CUX1* genes in samples mutated for *FLNB* and *TMEM132D* genes, respectively ($p < 0.008$). Furthermore, we compared the mutational status of these 60 genes with copy number variations (CNVs). Specifically, we compared the number of altered copies (gains or losses) between mutated and WT groups for these genes. We identified 12 genes whose mutational status was linked to copy number alterations in other genes

**Table 2 Comparison of expression level and mutational status for putative mini-driver genes.**

| Gene (expression levels) | Gene (mutational status) | p-value | FC (log$_{10}$) | Gene (expression levels) | Gene (mutational status) | p-value | FC (log$_{10}$) |
|---|---|---|---|---|---|---|---|
| CUX1 | TMEM132D | 0.0005 | −0.406 | ACVR2A | BMPR2 | 0.0028 | 0.567 |
| MYH11 | MYH11 | 0.0043 | −0.44 | DOCK3 | XIRP2 | 0.0028 | 0.58 |
| SYNE2 | LAMA2 | 0.0056 | 0.429 | MUC5B | COL7A1 | 0.0028 | 1.49 |
| UBR5 | COL7A1 | 0.006 | −0.456 | DOCK3 | SZT2 | 0.0027 | 1.143 |
| ZNF536 | FLNB | 0.0075 | −0.484 | MUC5B | LAMA2 | 0.0027 | 1.102 |
| FLNA | ACVR2A | 0.0075 | 0.463 | MUC5B | ATP10A | 0.0024 | 1.015 |
| FBN2 | FBN3 | 0.0079 | −0.4 | ACVR2A | ACVR2A | 0.0022 | 0.517 |
| DNAH7 | PCNT | 0.009 | 0.423 | MUC5B | ACVR2A | 0.0016 | 1.286 |
| SYNE2 | TRIOBP | 0.009 | 0.402 | MUC5B | ADGRV1 | 0.0013 | 1.003 |
| SYNE2 | SRRM2 | 0.0099 | 0.434 | UBR5 | FLNB | 0.0012 | −0.538 |
| DOCK3 | TCHH | 0.0095 | 0.608 | DOCK3 | FLNB | 0.0008 | 0.917 |
| DOCK3 | CELSR1 | 0.0085 | 0.846 | MUC5B | WDFY3 | 0.0007 | 1.077 |
| MUC5B | XIRP2 | 0.0084 | 1.107 | DOCK3 | MYO15A | 0.0006 | 0.979 |
| DOCK3 | ADGRV1 | 0.0081 | 0.541 | MUC5B | MYO15A | 0.0005 | 1.442 |
| DOCK3 | DNAH1 | 0.0074 | 0.608 | MUC5B | ASPM | 0.0004 | 1.263 |
| SYNE2 | HSPG2 | 0.0074 | 0.611 | MUC5B | TENM2 | 0.0003 | 1.271 |
| MUC5B | RP1L1 | 0.0074 | 1.426 | MUC5B | KIAA1109 | 0.0003 | 1.013 |
| ACVR2A | SIPA1L3 | 0.0072 | 0.5 | SYNE2 | ASPM | 0.0003 | 0.59 |
| DOCK3 | POLE | 0.0066 | 0.726 | MUC5B | FLNB | 0.0002 | 1.51 |
| MUC5B | BMPR2 | 0.004 | 1.559 | MUC5B | DNAH7 | 0.0002 | 1.494 |
| DOCK3 | BMPR2 | 0.0039 | 0.903 | MUC5B | POLE | 0.0001 | 1.373 |
| DOCK3 | TRPM3 | 0.0039 | 0.515 | | | | |

**Note:**
Genes shown log$_{10}$(FC) > 0.4 and $p < 0.01$. FC, Fold change.

($p < 0.05$, Table S3). Interestingly, the mutated *SLITRK5* gene cluster was related to copy number alterations in five genes (*NOTCH3, FBN3, SIPA1L3, PTPRS,* and *CUX1*), particularly in *NOTCH3* and *FBN3*, where the *SLITRK5* mutated group showed large ratios (greater than 4) compared to WT patients (Fig. 5).

Overall, our results demonstrate that *MUC5B, TMEM132D, SYNE2, POLE, CUX1,* and *NOTCH3* play critical roles as effectors with altered expression levels (Table S2) or copy number variations (Table S3) in the context of mini-driver genes.

## Mini-driver genes contribute to CRC prognosis

We used the 60 genes obtained to analyze their performance as biomarkers for survival in CRC (Fig. 3). We applied this analysis to 201 CRC patients that had complete information on overall survival rates and mutational profiles of the 60 genes. The mutational status of five genes, *DOCK3, FN1, PAPPA2, DNAH11,* and *FBN2*, were associated ($p < 0.05$) with poor survival in CRC patients (Fig. 6A). We observed a survival rate of ~25% or less for the mutated group, while the wild-type (WT) group retained more than 75% of survival after 5 years of follow-up. Then, when we constructed a panel with these genes, we observed an

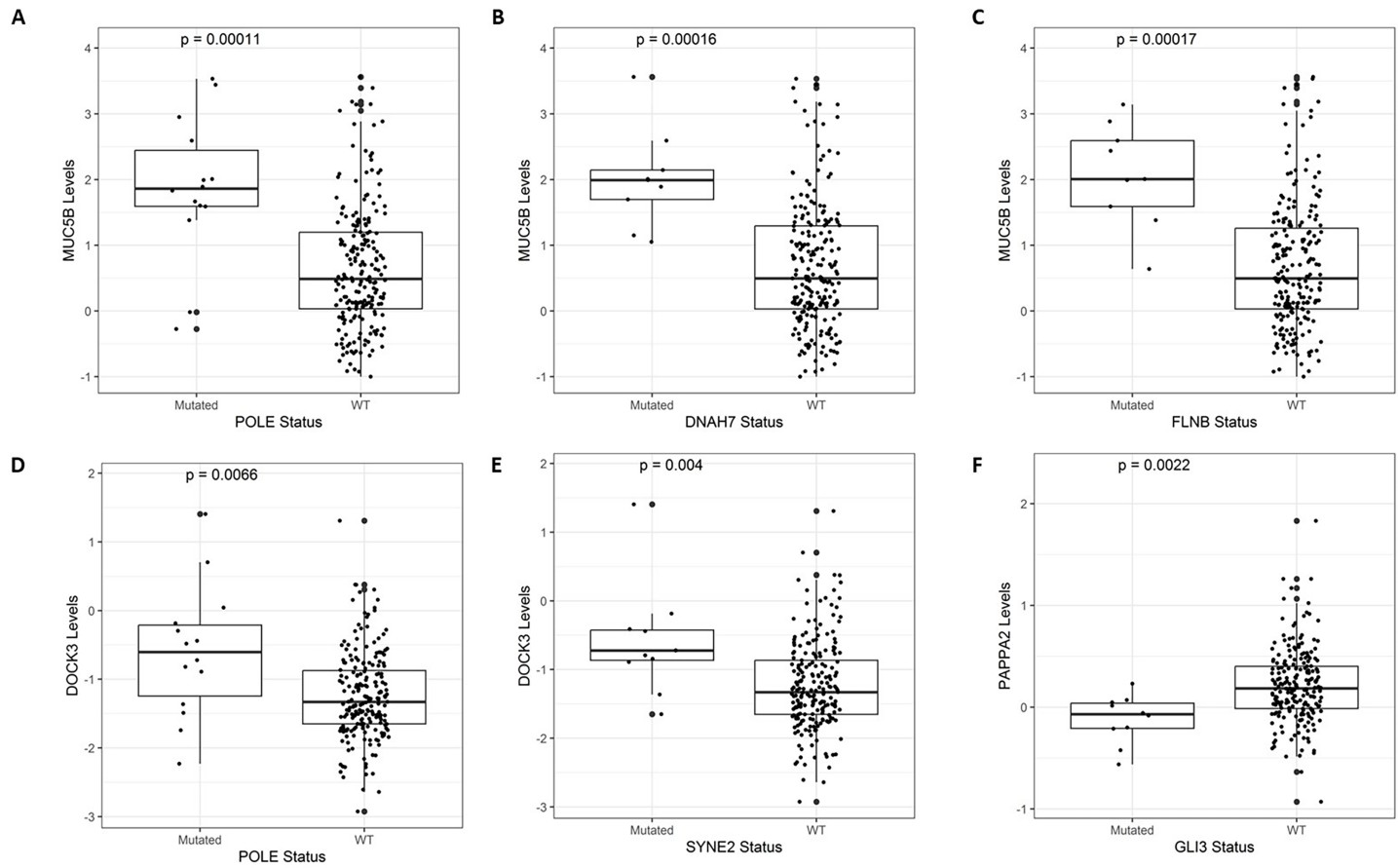

**Figure 4  A selection of genes whose mutational status was associated with alterations in expression levels.** (A) *MUC5B vs. POLE* (*p* = 0.00011). (B) *MUC5B vs. DNAH7* (*p* = 0.00016). (C) *MUC5B vs. FLNB* (*p* = 0.00017). (D) *DOCK3 vs. POLE* (*p* = 0.0066). (E) *DOCK3 vs. SYNE2* (*p* = 0.0056). (F) *PAPPA2 vs. GLI3* (*p* = 0.0022).                    

improvement in the survival rates differences between both groups (median $OS_{mutated}$ = 38.5 months *vs.* median $OS_{WT}$ = Not reached; Log-rank, *p* < 0.0001; Fig. 6B). Interestingly, we were able to identify a gene expression signature constituted of 16 genes that characterize the mutated group (*p* < 0.05 and absolute FC > 1.5). Between these genes, *DOCK3, FN1, ADAMTS2, AHNAK, AHNAK2, DNAH7, NBEA, SACS, SMAD4,* and *VWF*, were upregulated in the mutated group; whereas *AMER1, DIDO1, LRP1, LRP1B, RNF43,* and *TG*, were downregulated (Fig. S1). Finally, we tested the ability to predict the outcome of our mini-driver gene panel, in the presence of possible confounding variables (Fig. 6C). Our multivariate Cox regression demonstrated that our gene panel maintains its usefulness in predicting prognosis independent of age, sex, and AJCC tumoral staging (HR = 2.92, *p* = 0.002).

Finally, Fig. 7 presents a visual abstract to summarize all our findings in this study and how our strategy could propose mini-driver genes as an additional set of markers to support current driver genes in CRC.

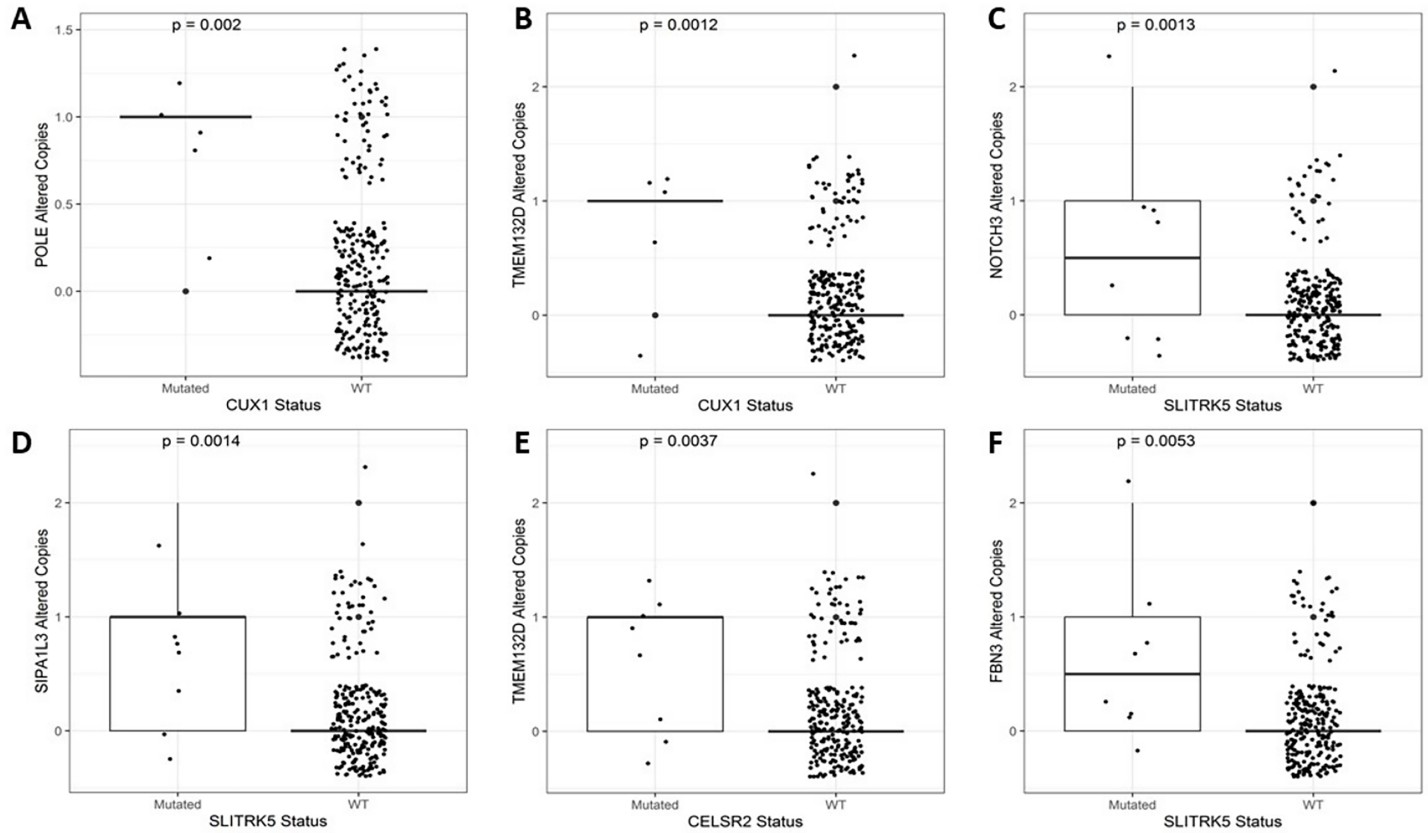

**Figure 5 A selection of genes whose mutational status was associated with alterations in the number of copies.** (A) *POLE vs. CUX1* (*p* = 0.002). (B) *TMEM132D vs. CUX1* (*p* = 0.0012). (C) *NOTCH3 vs. SLITRK5* (*p* = 0.0013). (D) *SIPA1L3 vs. SLITRK5* (*p* = 0.0014). (E) *TMEM132D vs. CELSR2* (*p* = 0.0037). (F) *FBN3 vs. SLITRK5* (*p* = 0.0053).

## DISCUSSION

In this study, we utilized three reliable databases of colorectal cancer patients to propose novel perspectives on the analysis of putative mini-driver genes. These studies provide crucial and representative information on colorectal cancer. For instance, DFCI's study focused on molecular characterization utilizing whole exome sequencing (WES) to gather tumor genomic data alongside detailed pathological and clinical information (*Giannakis et al., 2016*). TCGA's comprehensive analysis using around 10,000 samples representing 33 cancer types was employed for our interest in gene expression analysis corresponding to CRC samples (*Liu et al., 2018*). Finally, MSKCC's study analyzed inter- and intratumoral heterogeneity as evidence in the development of CRC (*Brannon et al., 2014*).

As mini-driver genes have a low mutational frequency, their impact alone is insufficient to generate a significant advantage for tumor cells. *Badr et al. (2022)* state that multiple accumulated weak mutations can combine to form a polygenic conductor (main driver) with enough impact to modify cellular function and patient prognosis (*Badr et al., 2022*), as depicted in Fig. 6. Our findings indicate that patients with mutations in *DOCK3*, *FN1*, *PAPPA2*, *DNAH11*, and *FBN2* genes had a shorter survival rate compared to patients without mutations. These results suggest that these alterations lead to cell dysregulation, as

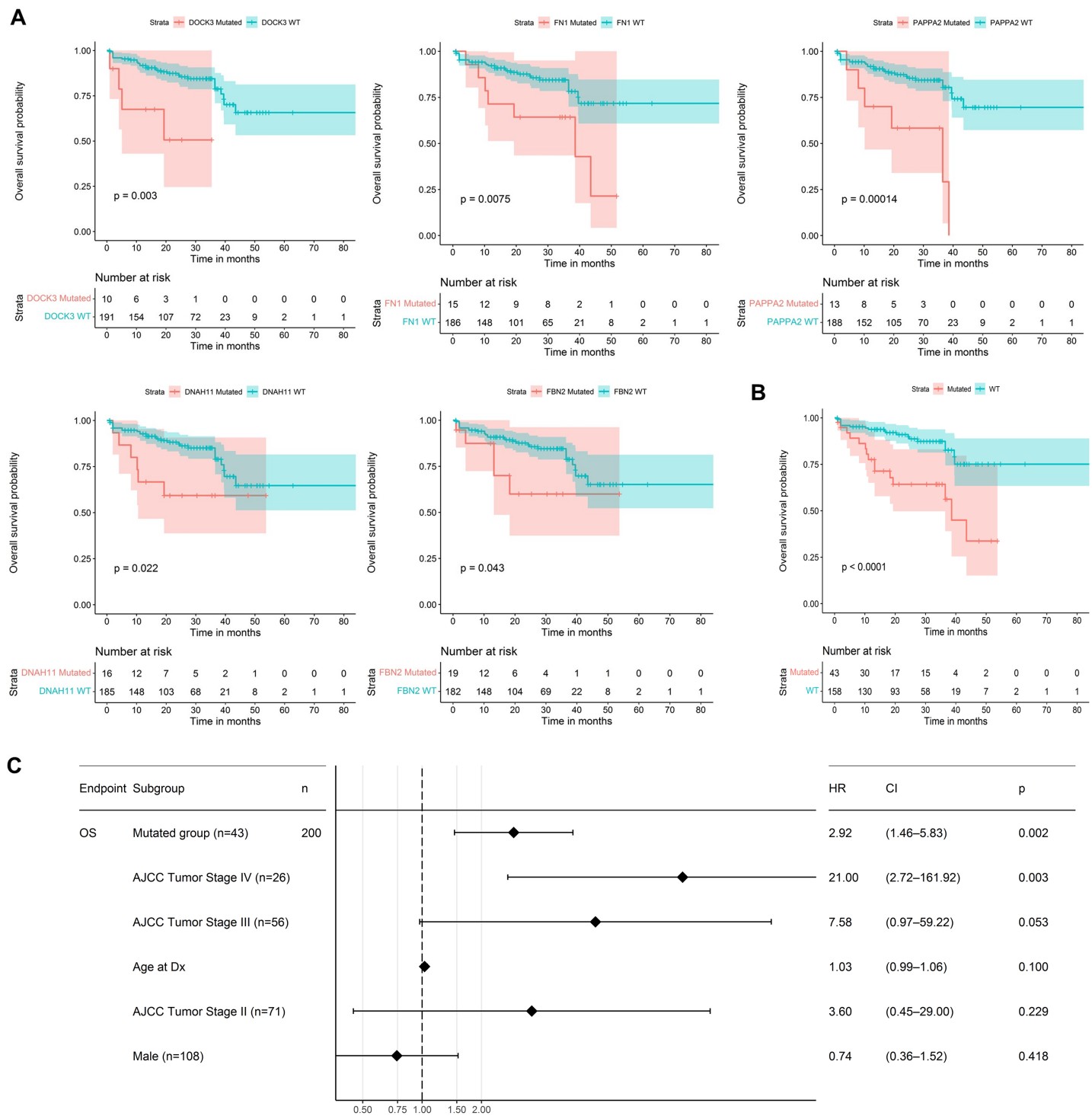

**Figure 6 Prognosis analysis of five potential mini-drivers.** (A) Kaplan–Meier curves analyses according to the mutational status of the *DOCK3, FN1, PAPPA2, DNAH11,* or *FBN2* genes. (B) Overall survival curve according to the combined mutational profile of *DOCK3, FN1, PAPPA2, DNAH11,* and *FBN2* genes. (C) Multivariate Cox regression demonstrating that the prognostic value of our gene panel is not included in variables commonly associated with the outcome. HR, Hazard ratio; CI, Confidence interval; *n*, number of samples.

**Features of Mini-driver genes in Colorectal Cancer Samples**

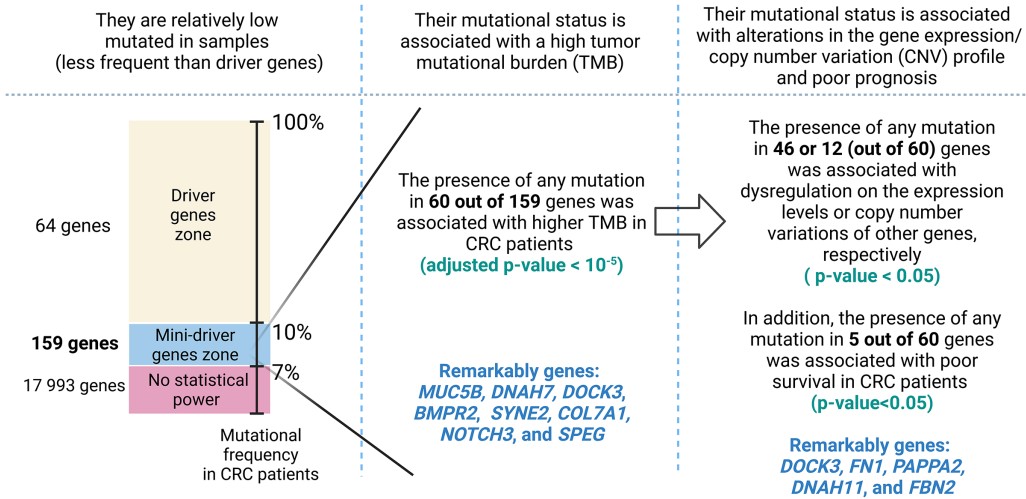

**Figure 7** **A visual abstract of the study.** Since mutations in driver genes use canonical criteria to be characterized and could be not sufficient to explain all cancer cases, we evaluated a strategy for proposing additional genes using the mini-driver hypothesis.

seen in other cancer types (*Irmak-Yazicioglu, 2016*; *Wilk & Braun, 2018*; *Furuya et al., 2021*).

The *DOCK3* gene has been reported to participate in various processes related to invasion, migration, and metastasis in cancer cells (*Hofer et al., 2017*; *Kotelevets & Chastre, 2020*; *Lu et al., 2020*). However, there is limited research on the involvement of the *MUC5B* gene in CRC, with recent evidence showing high expression of this gene in elderly CRC patients, particularly in poorly differentiated tumors (*Iranmanesh et al., 2021*). Extracellular vesicles with high levels of phosphorylated and expressed *FN1* have been identified as potential prognostic factors and therapeutic targets in CRC (*Qi et al., 2020*; *Zheng et al., 2020*). It has been postulated that *FN1* could function as a promoter gene in non-canonical pathways of mini-driver genes and their mutations, with upregulation of *FN1* by the *HMGA2* gene contributing to a metastatic profile in CRC cells (*Wu et al., 2016*). Mutations in the *PAPPA2* have been associated with tumor progression and treatment of digestive tumors, although its role in CRC is not yet understood (*Miao et al., 2022*). Similarly, the rs2285947 polymorphism in the *DNAH11* gene has been linked to an increased risk of several cancer types, suggesting its potential contribution as a mini-driver gene in carcinogenesis (*Wang et al., 2015*).

The *FBN2* gene is hypermethylated in CRC tissue and serum samples from patients with CRC and liver metastases, and its expression is directly correlated with shorter survival rates in colon cancer patients, suggesting a possible role as a tumor suppressor gene (*Leygo et al., 2017*; *Wang et al., 2022*). Although methylation data for all genes in a representative number of CRC samples were not available for this study, we anticipate that upcoming omics data will provide more information on methylation levels in CRC patients, allowing

us to update this perspective and stimulate more NGS experiments for a more robust comparison.

Other genes, such as *NOTCH3* and *SLITRK5*, have clear contributions to tumor development. *NOTCH3* promotes tumor cell survival and proliferation, induces EMT and cancer stem cell (CSC) properties, and has been linked to various clinical and pathological features, including larger tumor size, advanced TNM stage, higher pathological grade, and tumor metastasis (*Pastò et al., 2014*; *Aburjania et al., 2018*; *Xiu et al., 2021*). Frequent genetic, epigenetic, and transcriptional changes have been observed in the *SLITRK5* gene in colorectal neoplasias (*Hesson et al., 2016*). Nevertheless, genes such as *CUX1* have controversial reports. It has been recently discovered that *CUX1* is a tumor suppressor paradoxically overexpressed in tumor cells (*Cancer Genome Atlas Network, 2012*; *Jo et al., 2017*; *Liu et al., 2020*). Overall, a mini-driver gene approach could be a useful tool to support further analysis (sense and antisense strands) of these controversial regions to understand their involvement in tumor growth.

According to *Dressler et al. (2022)*, when a driver gene mutates, it significantly impacts cancer cell growth, providing them with a fundamental advantage in their development (*Dressler et al., 2022*). However, our classification of patients as mutated, which includes at least one mutation in one of five genes, has resulted in decreased survival time. The universe of mutations present in these genes may have different pathways to favor tumor proliferation. Nonetheless, the classification of somatic mutations is affected by the initial analysis. Typically, researchers detect all somatic mutations but exclude those highly present in large populations (*Timmermann et al., 2010*; *Ma et al., 2020*). This can limit the analysis to a reduced number of targets (*Leedham & Tomlinson, 2012*; *Lee-Six et al., 2019*) based on the pathological effect related to variations rarely present in healthy individuals. However, in cancer, aberrant expression levels and unexpected pathways (*Hanahan, 2022*) may support new functions for these polymorphisms that are usually discounted when found as somatic variations.

Therefore, our study suggests analyzing all somatic mutations to assess cancer prognosis, combining traditionally evaluated driver genes and mutations with additional tumor-promoting regions (mini-drivers). We believe that even silent mutations may have additional functions related to the expression of non-coding RNA expressed from the same genomic locations. *Li & Thirumalai (2016)* argue that when the main drivers lose their mutagenic capacity, mini-drivers help normalize the physical condition advantage of the drivers. Additionally, mini-drivers could confer a physical fitness advantage on cancer cells, especially when they accumulate during tumor progression due to stochastic genetic mutations (*Li & Thirumalai, 2016*). To increase the number of therapy proposals and biomarkers discovered for colorectal cancer (CRC), we suggest exploring candidate mini-driver regions in a gene-panel strategy.

We used NGS-based information to analyze somatic mutational profiles from a broader perspective to evaluate a new strategy for predicting biomarkers for CRC. However, our study has limitations. We have no control over the data collection, information, and permissions declared by the patients. Nonetheless, the databases used are supported and validated by accredited public and private health institutions. Furthermore, many of the

proposed biomarkers for CRC have limited specificity and require further experimental research. Nevertheless, our methodology uses critical and rigorous settings to determine putative mini-driver genes, providing deeper insights into colorectal carcinogenesis to mitigate the risk of presenting biased information. Therefore, we expect that our findings will support future research to find prognostic markers for CRC using solid and liquid biopsies by testing panels of driver and mini-driver genes.

### Funding

This work was supported by the Universidad Nacional Federico Villarreal (Lima, Peru) through Resolution No. 9343-2021-UNFV conceded to Anthony Vladimir Campos Segura and Prof. Ana Isabel Flor Gutiérrez Román as part of the incentives program for undergraduate thesis. The funders had no role in study design, data collection and analysis, decision to publish, or preparation of the manuscript.

### Grant Disclosures

The following grant information was disclosed by the authors:
Universidad Nacional Federico Villarreal (Lima, Peru): 9343-2021-UNFV.

### Competing Interests

The authors declare that they have no competing interests.

### Author Contributions

- Anthony Vladimir Campos Segura conceived and designed the experiments, performed the experiments, analyzed the data, prepared figures and/or tables, authored or reviewed drafts of the article, and approved the final draft.
- Mariana Belén Velásquez Sotomayor analyzed the data, prepared figures and/or tables, authored or reviewed drafts of the article, and approved the final draft.
- Ana Isabel Flor Gutiérrez Román performed the experiments, analyzed the data, authored or reviewed drafts of the article, and approved the final draft.
- César Alexander Ortiz Rojas analyzed the data, prepared figures and/or tables, authored or reviewed drafts of the article, and approved the final draft.
- Alexis Germán Murillo Carrasco conceived and designed the experiments, performed the experiments, analyzed the data, prepared figures and/or tables, authored or reviewed drafts of the article, and approved the final draft.

### Data Availability

The following datasets are available at cBioPortal (https://www.cbioportal.org/): Colorectal Adenocarcinoma (DFCI, Cell Reports 2016), Colorectal Adenocarcinoma (TCGA, PanCancer Atlas), and Colorectal Adenocarcinoma Triplets (MSK, Genome Biol 2014).
The mutations per gene and patient is available in the Supplemental File.

## Supplemental Information

Supplemental information for this article can be found online at http://dx.doi.org/10.7717/peerj.15410#supplemental-information.

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
