# Peer review of "Impact of mini-driver genes in the prognosis and tumor features of colorectal cancer samples: a novel perspective to support current biomarkers"

_PeerJ, doi:10.7717/peerj.15410_

## Round 0.1 · original submission · Major Revisions

· Academic Editor

Major Revisions

I would like to emphasize that the problem you are presenting is appealing, attractive, and important. However, I have to agree with Reviewer 2, in that you have not done a good job explaining your Experimental Rationale and/or making the necessary logical connections between the different steps in your experimental approach. More concerning to me is that you never formally defined what is what you are calling a "Mutation." Who calculated the Mutational Profile? How? Mutations can come in different flavors and will have different consequences. More importantly, you do not seem to have distinguished what is a normally present Single Nucleotide Polymorphism or InDel, associated with a given human population, versus the presence of a seriously deleterious mutation. What is what you call "Mutated" versus "Wild-Type"? In addition, you have not made a difference by sex or race or age. You claim to have incorporated in your analysis both genomic and transcriptomic data, yet it is totally unclear to me what data was downloaded and how that data was used. For example, I cannot tell from your description if you used Next Generation Genomic and/or Transcriptomic reads or numbers compiled by microarray hybridization. Did you use mRNA abundance values calculated by you or that were previously calculated? How was gene differential expression calculated? Also, you make no mention of how the biology of transcripts isoforms was included in your analysis. As presented, there is a disconnect between the objectives of your manuscript and your results, or at least how you are presenting your results.

If you decide to resubmit your manuscript to this journal, please make sure you incorporate and address all the different comments presented by the reviewers and by me.

Reviewer 1 ·

Basic reporting

This manuscript by Segura et al. aims to answer a question that has been understudied, what are the mini-driver genes in colorectal cancers (CRC) and how do they collectively impact the prognosis and tumor features? The paper is very well-written and the analytical logic was described properly. The background was sufficiently provided in the introduction such that extensive efforts have been made to study the driver genes but there are also other frequently mutated genes whose functions are currently unknown. With more experimental confirmation, this could provide the field with a novel way of studying mutations.

This paper generated no new datasets and used several established CRC datasets from cbioportal for analysis. Data sources are clearly given in the text.

Figures are adequately generated and all fonts are readable (the only comment would be for Figure 3, please adjust the positions of p-values in the box plots, and consider removing a few panels and only leave the most important findings in the main figures).

The Discussion section in its current form is too long. Maybe consider moving some of the description of gene sets to the Result section.

Experimental design

A few comments on the analysis:

1. What is the accurate definition of mini-drivers? Are these 7%, and 10% cutoffs used in other studies, or did the authors set them? The authors may want to justify the parameters they used to identify these mutated genes. For example, why not use 15% because the smallest driver gene mutation rate is 16%? The number of genes is significant in this analysis as it will decide the base number of mutated genes or mini-drivers therefore it needs to be adequately defined.

2. The authors used three datasets taken from cbioportal for mutation analysis. How did the authors identify the statistically significantly mutated genes? Did the authors use MutSigCV or any established packages? Please specify.

3. Other than mutations, are these mini-driver genes also showing copy number changes? Could the authors comment on that?

Validity of the findings

The findings in this manuscript is validated by survival analysis performed on the datasets. However, it is curious how the survival analysis would vary if the authors control for different clinical features, for example, age groups, ancestry groups or sex. Is it possible that these mini-driver genes affect one clinical phenotype more than the other?

·

Basic reporting

Figure-2 legend, is 10-5 a typo?

It is unclear, the criteria that can classify a set of genes as “mini drivers” ?

Line 170: “previously filtered genes were associated with high mutation rates”. What are these filtered genes? Are these genes under low mutation group (159 genes)?

Figure 2 needs more details. It is unclear what the authors want to achieve from the analysis. Please include statistical details that allow authors to associate low mutation group genes with high mutation rates.

Line 178: How can authors call genes with low mutation rate a ”list of potential mini-drivers”?

Line 180-181: “performed an analysis by comparing expression levels measured by the values of FC (log10). Then”, where are the results?

Line 183: “Gene pairs”, is it unclear which pair of genes authors are referring to?

Authors must specify the work association here, is it “correlation” or “regression” or “differential expression”? How did the authors associate a pair of genes? Authors must include the coefficient of the significance of the association.

Line 198: Where did the authors get the 60 genes for subsequent analysis? Please elaborate.

Line 200: Are these 5 genes mini-drivers? do they come under the low mutation rate group?

In the introduction, the authors clearly mentioned that these mini-drivers have weak tumor-promoting effects. It looks like, the workflow performs an integrative analysis of gene mutation rate and their expression profile. If these genes are differentially expressed and are directly affecting the survival rate of CRC patients, how can we call them “mini-drivers”?

Authors should enumerate the polygenic or combined effect of the shortlisted genes or mini-driver genes.

Experimental design

no comment

Validity of the findings

no comment

Additional comments

Experimental design and workflow are poorly presented. Authors need to streamline the results in a scientific manner. I found sentences unconnected and too confusing to read. The results must be redrafted while keeping the main hypothesis of the study (i.e. mini-drivers) in consideration.

---

## Round 0.2 · accepted · Accept

· Academic Editor

Accept

I am glad to inform you that your manuscript was found suitable for publication. I wanted to apologize for the long wait, but unfortunately, we were waiting on one of the reviewers to make a determination. Please make sure to correct any typos that might have escaped our review process. Feel free to contact me if you have any questions.

Reviewer 1 ·

Basic reporting

The authors have adequately addressed all previous comments. The revised manuscript now is much clearer in its description and easier to follow.

Experimental design

The authors have adequately addressed all previous comments.

Validity of the findings

The authors have adequately addressed all previous comments.